# Joint CTC-attention End-to-end Speech Recognition

## Abstract

End-to-end automatic speech recognition (ASR) has become a popular alternative to conventional DNN/HMM hybrid systems because it avoids the need for linguistic resources such as pronunciation dictionary, tokenization, and context-dependency trees, leading to a greatly simplified model-building process. There are two major types of end-to-end architectures for ASR: attention-based methods use an attention mechanism to perform alignment between acoustic frames and recognized symbols, and connectionist temporal classification (CTC), uses Markov assumptions to efficiently solve sequential problems by dynamic programming. This paper proposes joint CTC-attention end-to-end ASR, which effectively utilizes both advantages in training and decoding. We have applied the proposed method to two ASR benchmarks (spontaneous Japanese and Mandarin Chinese), and showing the comparable performance to conventional hybrid ASR systems without linguistic resources.

## 1 Introduction

Automatic speech recognition (ASR) is currently a mature set of technologies that have been widely deployed, resulting in great success in interface applications such as voice search. A typical ASR system is factorized into several modules including acoustic, lexicon, and language models based on a probabilistic noisy channel model (Jelinek, 1976). Over the last decade, dramatic improvements in acoustic and language models have been driven by machine learning techniques known as deep learning (Hinton et al., 2012). However,

current systems lean heavily on the scaffolding of complicated legacy architectures that grew up around traditional techniques. These present the following problems that we may seek to eliminate.

1. Flat start: many module-specific processes are required to build an accurate module: for example, when we build an acoustic model from scratch, we have to first build hidden Markov model (HMM) and Gaussian mixture model (GMM) followed by deep neural networks (DNN).

2. Linguistic knowledge: to well factorize acoustic and language models, we need to have a lexicon model, which is usually based on a hand-crafted pronunciation dictionary to map word to phoneme sequence. Also, some languages do not explicitly have a word boundary and need tokenization modules (Kudo et al., 2004; Bird, 2006).

3. Conditional independence assumptions: the current ASR systems often use conditional independence assumptions (especially Markov assumptions) during the above factorization and to make use of GMM, DNN, and $n$-gram models. The data do not necessarily follow such assumptions leading to model mis-specification.

4. Complex decoding: inference/decoding has to be performed by integrating all modules. Although this integration is often efficiently handled by finite state transducers, the construction and implementation of well-optimized transducers is very complicated.

5. Local optimum: the above modules are optimized separately, which may result in local optima, where each module is trained to match the other modules[1].

---

[1] Sequence discriminative training (Veselý et al., 2013) can solve the issue to some extent, but requires additional

Consequently, it is quite difficult for non-experts to use/develop ASR systems for new applications, especially for new languages.

End-to-end ASR has the goal of simplifying the above module-based architecture into a single-network architecture within a deep learning framework, in order to address the above issues. There are two major types of end-to-end architectures for ASR: attention-based methods use an attention mechanism to perform alignment between acoustic frames and recognized symbols, and connectionist temporal classification (CTC), uses Markov assumptions to efficiently solve sequential problems by dynamic programming (Chorowski et al., 2014; Graves and Jaitly, 2014).

All ASR models aim to elucidate the posterior distribution $p(W|X)$ of word sequence $W$ given speech feature sequence $X$. The end-to-end methods do this directly whereas conventional models factorize $p(W|X)$ into modules such as the language model $p(W)$, which can be trained on pure language data, and an acoustic model likelihood $p(X|W)$ which is trained on acoustic data with the corresponding language labels. End-to-end ASR methods typically rely only on paired acoustic and language data. Without the additional language data, they can suffer from data sparseness or out-of-vocabulary issues. To improve generalization, and handle out-of-vocabulary problems, it is typical to use the letter representation rather than the word representation for the language output sequence, which we adopt in the descriptions below.

The attention-based end-to-end method solves the ASR problem as a sequence mapping from speech feature sequences to text by using encoder-decoder architecture. The decoder network uses an attention mechanism to find an alignment between each element of the output sequence and the hidden states generated by the acoustic encoder network for each frame of acoustic input (Chorowski et al., 2014, 2015; Chan et al., 2015; Lu et al., 2016). At each output position, the decoder network computes a matching score between its hidden state and the states of the encoder network at each input time, to form a temporal alignment distribution, which is then used to extract an average of the corresponding encoder states.

This basic temporal attention mechanism is too flexible in the sense that it allows extremely non-

———————————

process in the above modules including lattice generations.

sequential alignments. This may be fine for applications such as machine translation where input and output word order are different (Bahdanau et al., 2014; Wu et al., 2016). However in speech recognition, the feature inputs and corresponding letter outputs generally proceed in the same order, with only small within-word deviations (e.g. the word "iron" which transposes the sounds for "r" and "o"). Another problem is that the input and output sequences in ASR can have very different lengths, and these vary greatly from case to case, depending on the speaking rate and writing system, making it more difficult to track the alignment.

However, an advantage is that the attention mechanism does not require any conditional independence assumptions, and could address all the problems cited above. Although the alignment problems of attention-based mechanisms have been partially addressed in (Chorowski et al., 2014; Chorowski and Jaitly, 2016) using various mechanisms, here we propose more rigorous constraints by using CTC-based alignment to guide the training.

CTC permits an efficient computation of a strictly monotonic alignment using dynamic programming (Graves et al., 2006; Graves and Jaitly, 2014) although it requires language models and graph-based decoding (Miao et al., 2015) except in the case of huge training data (Amodei et al., 2015; Soltau et al., 2016). We propose to take advantage of the constrained CTC alignment in a hybrid CTC-attention based system. During training, we attach a CTC objective to an attention-based encoder network as a regularization, as proposed by (Kim et al., 2016). This greatly reduces irregular alignments without any heuristic search techniques. During decoding, we propose to use a rescoring technique, where hypotheses of attention-based ASR are refined by scores obtained by using encoder outputs.

The proposed method is applied to Japanese and Mandarin ASR tasks, which require extra linguistic resources including morphological analyzer (Kudo et al., 2004) or word segmentation (Xue et al., 2003) in addition to pronunciation dictionary to provide accurate lexicon and language models in conventional hybrid ASR. Surprisingly, the method achieved performance comparable to, and in some cases superior to, several state-of-the-art hybrid ASR systems, without using the above

linguistic resources.

## 2 From hybrid to end-to-end ASR

This section briefly provides a formulation of conventional HMM/DNN hybrid ASR and CTC or attention based end-to-end ASR. The formulation is intended to clarify the conditional independence assumption (Markov assumption), which is an important property to characterize these three methods.

### 2.1 Hybrid HMM/DNN

ASR deals with a sequence mapping from $T$-length speech feature sequence $X = \{\mathbf{x}_t \in \mathbb{R}^D | t = 1, \cdots, T\}$ to $N$-length word sequence $W = \{w_n \in \mathcal{V} | n = 1, \cdots, N\}$. $\mathbf{x}_t$ is a $D$ dimensional speech feature vector (e.g., log Mel filterbanks) at frame $t$ and $w_n$ is a word at position $n$ in vocabulary $\mathcal{V}$.

ASR is mathematically formulated with the Bayes decision theory, where the most probable word sequence $\hat{W}$ is estimated among all possible word sequences $\mathcal{V}*$ as follows:

$$\hat{W} = \arg \max_{W \in \mathcal{V}*} p(W|X). \qquad (1)$$

Therefore, the main problem of ASR is how to obtain the posterior distribution $p(W|X)$.

In the current main stream of ASR is called hybrid HMM/DNN (Bourlard and Morgan, 1994), which uses the Bayes theorem and introduces HMM state sequence $S = \{s_t \in \{1, \cdots, J\} | t = 1, \cdots, T\}$ to factorize $p(W|X)$ into the following three distributions:

$$\arg \max_W p(W|X)$$
$$\approx \arg \max_W \sum_S p(X|S)p(S|W)p(W). \quad (2)$$

The three factors, $p(X|S)$, $p(S|W)$, and $p(W)$, are acoustic, lexicon, and language models, respectively. Eq. (2) is obtained by a conditional independence assumption (i.e., $p(X|S, W) \approx p(X|S)$), which is a reasonable assumption to simplify the dependency of the acoustic model.

**Acoustic model** $p(X|S)$

$p(X|S)$ is further factorized by using a probabilistic chain rule and conditional independence assumption as follows:

$$p(X|S) \approx \prod_t p(\mathbf{x}_t|s_t) \propto \prod_t \frac{p(s_t|\mathbf{x}_t)}{p(s_t)}, \quad (3)$$

where the framewise likelihood function $p(\mathbf{x}_t|s_t)$ is replaced with the framewise posterior distribution $p(s_t|\mathbf{x}_t)/p(s_t)$ computed by powerful DNN classifiers by using so-called pseudo likelihood trick (Bourlard and Morgan, 1994). The conditional independence assumption in Eq. (3) is often regarded as too strong assumption, since it does not consider any input and hidden state contexts. Therefore DNNs with long context features or recurrent neural networks are often used to mitigate the issue. To train the framewise posterior, we also require to provide a framewise state alignment $s_t$ as a target, which is often provided by a HMM/GMM system.

**Lexicon model** $p(S|W)$

$p(S|W)$ is also factorized by using a probabilistic chain rule and conditional independence assumption (1st-order Markov assumption) as follows:

$$p(S|W) \approx \prod_t p(s_t|s_{t-1}, W) \qquad (4)$$

This probability is represented by an HMM state transition given $W$. The conversion from $W$ to HMM states is deterministically performed by using a pronunciation dictionary through a phoneme representation.

**Language model** $p(W)$

Similarly, $p(W)$ is factorized by using a probabilistic chain rule and conditional independence assumption ($m - $ 1th-order Markov assumption) as a $m$-gram model, i.e., $p(W) \approx \prod_n p(w_n|w_{n-1}, \ldots, w_{n-m-1})$. Although recurrent neural network language models (RNNLMs) can avoid this conditional independence assumption issue (Mikolov et al., 2010), it makes the decoding complex, and RNNLMs are often combined with $m$-gram language models based on as a re-scoring technique.

Thus, conventional hybrid HMM/DNN systems make the ASR problem formulated with Eq. (1) feasible by using factorization and conditional independence assumptions, at the cost of the five problems discussed in Section 1.

### 2.2 Connectionist Temporal Classification (CTC)

The CTC formulation also follows from Bayes decision theory (Eq. (1)). Note that the CTC formulation uses $L$-length letter sequence $C = \{c_l \in \mathcal{U} | l = 1, \cdots, L\}$ with a set of distinct letters $\mathcal{U}$.

Similarly to Section 2.1, by introducing framewise letter sequence with an additional "blank" symbol $Z = \{z_t \in \mathcal{U} \cup \text{blank} | t = 1, \cdots, T\}$, and by using the probabilistic chain rule and conditional independence assumption, the posterior distribution $p(C|X)$ is factorized as follows:

$$p(C|X)$$
$$\approx \sum_Z p(C|Z)p(Z|X) \approx \sum_Z p(C|Z) \prod_t p(z_t|X)$$
$$\approx \underbrace{\sum_Z \prod_t p(z_t|z_{t-1}, C)p(z_t|X)\, p(C)}_{\triangleq p_{\text{ctc}}(C|X)} \quad (5)$$

CTC is similar to the hybrid approach, except that it applies Bayes theorem to $p(C|Z)$ instead of to $p(W|X)$. As a result, CTC has three distribution components similar to the hybrid case, i.e., framewise posterior distribution $p(z_t|X)$, transition probability $p(z_t|z_{t-1}, C)$, and letter-based language model $p(C)$. We also define the CTC objective function $p_{\text{ctc}}(C|X)$ used in the later formulation.

The framewise posterior distribution $p(z_t|X)$ is conditioned on all inputs $X$, and it is quite natural to be modeled by using bidirectional long short-term memory (BLSTM):

$$p(z_t|X) = \text{Softmax}(\text{Lin}(\mathbf{h}_t)) \quad (6)$$
$$\mathbf{h}_t = \text{BLSTM}(X). \quad (7)$$

Softmax($\cdot$) is a sofmax activation function, and Lin($\cdot$) is a linear layer to convert hidden vector $\mathbf{h}_t$ to a $(|\mathcal{U}| + 1)$ dimensional vector (+1 means a blank symbol introduced in CTC).

Although Eq. (5) has to deal with a summation over all possible $Z$, it is efficiently computed by using dynamic programming (Viterbi/forward-backward algorithm) thanks to the Markov property. In summary, although CTC and hybrid systems are similar to each other due to conditional independence assumptions, CTC does not require pronunciation dictionaries and omits an HMM/GMM construction step.

### 2.3 Attention mechanism

Compared with hybrid and CTC approaches, the attention-based approach does not make any conditional independence assumptions, and directly estimates the posterior $p(C|X)$ based on a prob-

abilistic chain rule, as follows:

$$p(C|X) = \underbrace{\prod_l p(c_l|c_1, \cdots, c_{l-1}, X)}_{\triangleq p_{\text{att}}(C|X)}, \quad (8)$$

where $p_{\text{att}}(C|X)$ is an attention-based objective function. $p(c_l|c_1, \cdots, c_{l-1}, X)$ is obtained by

$$p(c_l|c_1, \cdots, c_{l-1}, X) = \text{Decoder}(\mathbf{r}_l, \mathbf{q}_{l-1}, c_{l-1})$$
$$\mathbf{h}_t = \text{Encoder}(X) \quad (9)$$
$$a_{lt} = \text{Attention}(\{a_{l-1}\}_t, \mathbf{q}_{l-1}, \mathbf{h}_t) \quad (10)$$
$$\mathbf{r}_l = \sum_t a_{lt}\mathbf{h}_t. \quad (11)$$

Eq. (9) converts input feature vectors $X$ into a framewise hidden vector $\mathbf{h}_t$ in an encoder network based on BLSTM, i.e., $\text{Encoder}(X) \triangleq \text{BLSTM}(X)$. Attention($\cdot$) in Eq. (10) is based on a content-based attention mechanism with convolutional features, as described in (Chorowski et al., 2015) (see Appendix B). $a_{lt}$ is an attention weight, and represents a soft alignment of hidden vector $\mathbf{h}_t$ for each output $c_l$ based on the weighted summation of hidden vectors to form letter-wise hidden vector $\mathbf{r}_l$ in Eq. (11). A decoder network is another recurrent network conditioned on previous output $c_{l-1}$ and hidden vector $\mathbf{q}_{l-1}$, similar to RNNLM, in addition to letter-wise hidden vector $\mathbf{r}_l$. We use Decoder($\cdot$) $\triangleq$ Softmax(Lin(LSTM($\cdot$))).

Attention-based ASR does not explicitly separate each module, and potentially handles the all five issues pointed out in Section 1. It implicitly combines acoustic models, lexicon, and language models as encoder, attention, and decoder networks, which can be jointly trained as a single deep neural network.

Compared with hybrid and CTC, which are based on a transition form from $t - 1$ to $t$ due to the Markov assumption, the attention mechanism does not maintain this constraint, and often provides irregular alignments. A major focus of this paper is to address this problem by proposing joint CTC-attention models.

## 3 Joint CTC-attention

This section explains a joint CTC-attention network, which utilizes both benefits of CTC and attention during training and decoding steps in ASR.

### 3.1 Multi-task learning

Kim et al. (2016) uses a CTC objective function as an auxiliary task to train the attention model en-

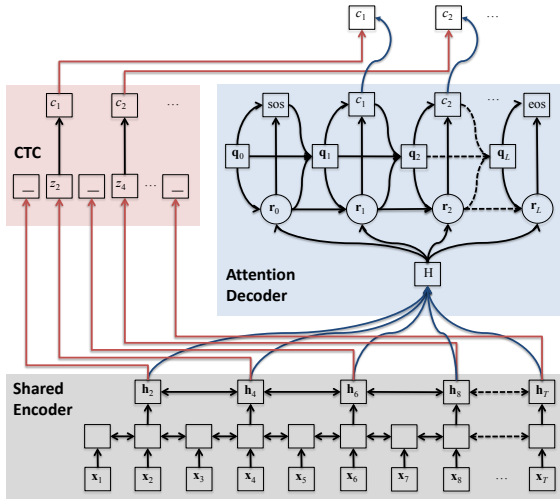

Figure 1: Joint CTC-attention based end-to-end framework: the shared encoder is trained by both CTC and attention model objectives simultaneously. The shared encoder transforms our input sequence $\{\mathbf{x}_t \cdots \mathbf{x}_T\}$ into high level features $H = \{\mathbf{h}_t \cdots \mathbf{h}_T\}$, and the attention decoder generates the letter sequence $\{c_1 \cdots c_L\}$.

coder within the multi-task learning (MTL) framework. Figure 1 illustrates the overall architecture of the framework, where the same BLSTM is shared with CTC and attention encoder networks (that is Eqs. (7) and (9), respectively). Unlike the sole attention model, the forward-backward algorithm of CTC can enforce monotonic alignment between speech and label sequences during training. That is, rather than solely depending on data-driven attention methods to estimate the desired alignments in long sequences, the forward-backward algorithm in CTC helps to speed up the process of estimating the desired alignment. The objective to be maximized is a logarithmic linear combination of the CTC and attention objectives, i.e., $p_{\text{ctc}}(C|X)$ in Eq. (5) and $p_{\text{att}}(C|X)$ in Eq. (8):

$$\mathcal{L}_{\text{MTL}} = \lambda \log p_{\text{ctc}}(C|X) + (1 - \lambda) \log p_{\text{att}}(C|X), \quad (12)$$

with a tunable parameter $\lambda : 0 \leq \lambda \leq 1$.

## 3.2 Joint CTC-attention decoding

The inference step of our joint CTC-attention based end-to-end speech recognition is performed by output-label synchronous decoding with a beam search like attention-based ASR. But, we take the CTC probabilities into account to find a better aligned hypothesis to the input speech, as shown in Figure 1

End-to-end speech recognition inference is generally defined as a problem to find the most probable letter sequence $\hat{C}$ given speech input $X$, i.e.

$$\hat{C} = \arg \max_{C \in \mathcal{U}^*} \log p(C|X). \quad (13)$$

In attention-based ASR, $p(C|X)$ is computed by Eq. (8), and $\hat{C}$ is found by the beam search.

Let $\Omega_l$ be a set of partial hypotheses of length $l$. At the beginning of the beam search, $\Omega_0$ contains only one hypothesis with starting symbol <sos>. For $l = 1$ to $L_{max}$, each partial hypothesis in $\Omega_{l-1}$ is expanded by appending possible single letters, and the new hypotheses are stored in $\Omega_l$, where $L_{max}$ is the maximum length of hypotheses to be searched. The score of each new hypothesis is computed in log domain as

$$\alpha(g_{l-1} \cdot c) = \alpha(g_{l-1}) + \log p(c|g_{l-1}, X), \quad (14)$$

where $g_{l-1}$ is a partial hypothesis in $\Omega_{l-1}$ and $c$ is a letter appended to $g_{l-1}$. The new partial hypothesis $g_l$, i.e. $g_{l-1} \cdot c$, is added to $\Omega_l$. If $c$ is a special symbol that represents the end of sequence, <eos>, $g_l$ is added to $\hat{\Omega}$ instead of $\Omega_l$, where $\hat{\Omega}$ denotes a set of complete hypotheses. Finally, $\hat{C}$ is obtained as

$$\hat{C} = \arg \max_{g \in \hat{\Omega}} \alpha(g). \quad (15)$$

In the beam search, $\Omega_l$ is allowed to hold only a limited number of hypotheses with higher scores to improve the search efficiency.

Attention-based ASR, however, may be prone to deletion and insertion errors (see Appendix A) because of its flexible alignment property, which can attend any portion of the encoder state sequence to predict the next label, as discussed in Section 2.3. Since the attention is generated by the decoder network it may prematurely predict the end-of-sequence label, even when it has not attended all the encoder frames, making the hypothesis too short. On the other hand, it may predict the next label with a high probability by attending the same portions as those attended before. In this case, the hypothesis becomes very long and includes repetitions of the same label sequence.

To alleviate this problem, a length penalty term is commonly used to control the hypothesis length to be selected (Chorowski et al., 2015; Bahdanau et al., 2016). With the length penalty, the decoding objective is changed to

$$\hat{C} = \arg \max_{C \in \mathcal{U}^*} \left\{ \log p(C|X) + \gamma |C| \right\}, \quad (16)$$

where $|C|$ is the length of sequence $C$ and $\gamma$ is a tunable parameter. However, it is actually difficult to completely exclude too short and too long hypotheses even if $\gamma$ is carefully tuned. It is also effective to control the hypothesis length by minimum and maximum lengths to some extent, where the minimum and the maximum are selected as fixed rations to the length of input speech. But, since there are exceptionally long or short transcripts compared to the input speech, it is difficult to balance saving such exceptional transcripts and preventing hypotheses with irrelevant lengths.

Another approach is a *coverage* term recently proposed in (Chorowski and Jaitly, 2016), which is incorporated in the decoding objective as

$$\hat{C} = \arg\max_{C \in \mathcal{U}^*} \{ \log p(C|X) + \gamma |C|$$
$$+ \eta \cdot \text{coverage}(C|X)\}, \quad (17)$$

and computed by

$$\text{coverage}(C|X) = \sum_{t=1}^{T} \left[ \sum_{l=1}^{L} a_{lt} > \tau \right], \quad (18)$$

where $\eta$ and $\tau$ are tunable parameters. The coverage term represents the number of frames that have received a cumulative attention greater than $\tau$. Accordingly, it increases when paying close attention to some frames for the first time, but does not increase when paying attention again to the same frames. This property is effective to avoid looping the same label sequence within a hypothesis. However, it is still difficult to obtain a common parameter setting for $\gamma, \eta, \tau$ and optional min/max lengths so that they are appropriate for any speech data from different tasks.

Our joint CTC-attention approach combines CTC and attention-based sequence probabilities in the inference step, as well as the training step. Suppose $p_{\text{ctc}}(C|X)$ in Eq. (5) and $p_{\text{att}}(C|X)$ in Eq. (8) are the sequence probabilities given by CTC and the attention model, respectively. The decoding objective is defined similarly to Eq. (12) as

$$\hat{C} = \arg\max_{C \in \mathcal{U}^*} \{ \lambda \log p_{\text{ctc}}(C|X)$$
$$+ (1 - \lambda) \log p_{\text{att}}(C|X)\}. \quad (19)$$

The CTC probability enforces a monotonic alignment that does not allow big jumps or looping the same frames. Accordingly, it is possible to choose a hypothesis with a better alignment and exclude irrelevant hypotheses without relying on any of the coverage term, the length penalty, and the min/max lengths.

We implement the joint CTC-attention inference method as a two-pass strategy. The first pass obtains a set of complete hypotheses using the beam search, where only attention-based sequence probabilities are considered. The second pass rescores the complete hypotheses using the CTC-attention probabilities, where the CTC probabilities are obtained by the forward algorithm for CTC (Graves et al., 2006). The rescoring pass obtains the final result according to

$$\hat{C} = \arg\max_{g \in \hat{\Omega}} \{ (1 - \lambda)\alpha(g) + \lambda\xi(g)\}, \quad (20)$$

where $\xi(g) = \log p_{\text{ctc}}(g|X)$.

# 4 Experiments

We used Japanese and Mandarin Chinese ASR benchmarks to show the effectiveness of the proposed joint CTC-attention approach. The main reason for choosing these two languages is that those ideogram languages have relatively shorter lengths for letter sequences than those in alphabet languages, which reduces computational complexities greatly, and makes it easy to handle context information in a decoder network. Our preliminary investigation shows that Japanese and Mandarin Chinese end-to-end ASR can be easily scaled up, and shows state-of-the-art performance without using various tricks developed in English tasks.

## 4.1 Corpus of Spontaneous Japanese (CSJ)

We demonstrated ASR experiments by using the Corpus of Spontaneous Japanese (CSJ) (Maekawa et al., 2000). CSJ is a standard Japanese ASR task based on a collection of monologue speech data including academic lectures and simulated presentations. It has a total of 581 hours of training data and three types of evaluation data, where each evaluation task consists of 10 lectures (totally 5 hours). As input features, we used 40 mel-scale filterbank coefficients, with their first and second order temporal derivatives to obtain a total of 120-dimensional feature vector per frame. The encoder was a 4-layer BLSTM with 320 cells in each layer and direction, and linear projection layer is followed by each BLSTM layer. The

Table 1: Character error rate (CER) for conventional attention and proposed joint CTC-attention end-to-end ASR. Corpus of Spontaneous Japanese speech recognition (CSJ) task.

| Model | Hour | Task1 | Task2 | Task3 |
|---|---|---|---|---|
| Attention | 581 | 11.4 | 7.9 | 9.0 |
| MTL | 581 | 10.5 | 7.6 | 8.3 |
| MTL + joint decoding | 581 | 10.1 | 7.1 | 7.8 |
| MTL-large + joint decoding | 581 | **9.4** | **7.0** | **7.5** |
| GMM-discr. (Moriya et al., 2015) | 236 for AM, 581 for LM | 11.2 | 9.2 | 12.1 |
| DNN-hybrid (Moriya et al., 2015) | 236 for AM, 581 for LM | 9.0 | 7.2 | 9.6 |
| CTC-syllable (Kanda et al., 2016) | 581 | 9.4 | 7.3 | 7.5 |

2nd and 3rd bottom layers of the encoder read every second hidden state in the network below, reducing the utterance length by the factor of 4. We used the content-based attention mechanism (Chorowski et al., 2015), where the 10 centered convolution filters of width 100 were used to extract the convolutional features. The decoder network was a 1-layer LSTM with 320 cells. The AdaDelta algorithm (Zeiler, 2012) with gradient clipping (Pascanu et al., 2012) was used for the optimization. The joint CTC-attention ASR was implemented by using the Chainer deep learning toolkit (Tokui et al., 2015).

Table 1 first compares the character error rate (CER) for conventional attention and MTL based end-to-end ASR without the joint decoding. $\lambda$ in Eq. (12) was set to 0.1. When decoding, we manually set the minimum and maximum lengths of output sequences by 0.1 and 0.5 times input sequence lengths, respectively. The length penalty $\gamma$ in Eq. (16) was set to 0.1. MTL significantly outperformed attention-based ASR in the all evaluation tasks, which confirms the effectiveness of joint CTC-attention model. Table 1 also shows that the joint decoding, described in Section 3.2, further improved the performance without setting any search parameters (maximum and minimum lengths, length penalty), but only setting a weight parameter $\lambda = 0.1$ in Eq. (20) similar to the MTL case. Figure 2 also compares the dependency of $\lambda$ on the CER for the CSJ evaluation tasks, and showing that $\lambda$ was not so sensitive to the performance if we set $\lambda$ around the value we used at MTL (i.e., 0.1).

We also compare the performance of the proposed MTL-large, which has a larger network (5-layer encoder network), with the conventional state-of-the-art techniques obtained by using linguistic resources. The state-of-the-art CERs of GMM discriminative training and DNN-sMBR

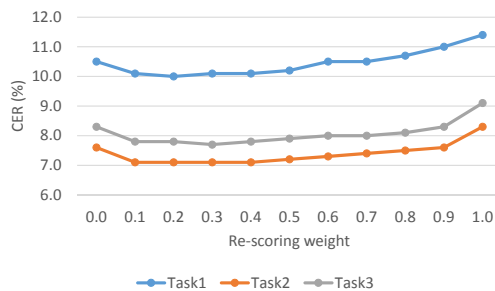

Figure 2: The effect of re-scoring weight $\lambda$ in Eq. (20) on the CSJ evaluation tasks.

hybrid systems are obtained from the Kaldi recipe (Moriya et al., 2015) and a system based on syllable-based CTC with MAP decoding (Kanda et al., 2016). The Kaldi recipe systems use academic lectures (236h) for AM training and all training-data transcriptions for LM training. Unlike the proposed method, these methods use linguistic resources including a morphological analyzer, pronunciation dictionary, and language model. Note that since the amount of training data and experimental configurations of the proposed and reference methods are different, it is difficult to compare the performance listed in the table directly. However, since the CERs of the proposed method are comparable to those of the best reference results, we can state that the proposed method reaches the state-of-the-art performance.

## 4.2 Mandarin telephone speech

We demonstrated ASR experiments on HKUST Mandarin Chinese conversational telephone speech recognition (MTS) (Liu et al., 2006). It has 5 hours recording for evaluation, and we extracted 5 hours from training data as a development set, and used the rest (167 hours) as a training set. All experimental conditions were same as those in Section 4.1 except that we used the $\lambda = 0.5$ in training and decoding instead of

Table 2: Character error rate (CER) for conventional attention and proposed joint CTC-attention end-to-end ASR. HKUST Mandarin Chinese conversational telephone speech recognition (MTS) task.

| Model | dev | eval |
|---|---|---|
| Attention | 40.3 | 37.8 |
| MTL | 38.7 | 36.6 |
| Attention + coverage | 39.4 | 37.6 |
| MTL + coverage | 36.9 | 35.3 |
| MTL + joint decoding | 35.9 | 34.2 |
| MTL + joint decoding (speed perturb.) | **32.1** | **31.4** |
| DNN-hybrid | – | 35.9 |
| LSTM-hybrid (speed perturb.) | – | 33.5 |
| CTC with language model (Miao et al., 2016) | – | 34.8 |
| TDNN-hybrid, lattice-free MMI (speed perturb.) (Povey et al., 2016) | – | 28.2 |

0.1 based on our preliminary investigation and 80 mel-scale filterbank coefficients with pitch feature as an input by referring (Miao et al., 2016). In decoding, we also added a result of the coverage-term based decoding (Chorowski and Jaitly, 2016), as discussed in Section 3.2 ($\eta = 1.5, \tau = 0.5, \gamma = -0.6$ for attention model and $\eta = 1.0, \tau = 0.5, \gamma = -0.1$ for MTL), since it was difficult to eliminate the irregular alignments during decoding by only tuning the maximum and minimum lengths and length penalty (we set the minimum and maximum lengths of output sequences by 0.0 and 0.4 times input sequence lengths, respectively and set $\gamma = 0.6$ in Table 2).

Table 2 shows the effectiveness of MTL and joint decoding over the attention-based approach, especially showing the significant improvement of the joint CTC-attention decoding. Similar to the CSJ experiments in Section 4.1, we did not use the length-penalty term or the coverage term in joint decoding. This is an advantage of joint decoding over conventional approaches that require many tuning parameters. We also generated more training data by linearly scaling the audio lengths by factors of 0.9 and 1.1 (speed perturb.). The final model achieved **31.4**%without using linguistic resources, which defeats moderate state-of-the-art systems including CTC-based methods[2].

---
[2]Although the proposed method did not reach the performance obtained by time delayed neural network (TDNN) with the lattice-free sequence discriminative training, this method fully utilizes linguistic knowledge through phoneme representation in the discriminative training (Povey et al., 2016), and it is hard to beat it without such knowledge.

## 5 Summary and discussion

This paper proposes end-to-end ASR by using joint CTC-attention, which outperformed attention-based end-to-end ASR by solving the misalignment issues. This method does not require the use of use linguistic resources including morphological analyzer, pronunciation dictionary, and language model, which are essential component of building conventional Japanese and Mandarin Chinese ASR systems. Nevertheless, the method achieved comparable performance to the state-of-the-art conventional systems for the CSJ and MTS tasks. In addition, the proposed method does not require GMM-HMM construction for initial alignments, DNN pre-training, lattice generation for sequence discriminative training, complex search in decoding (e.g., FST decoder or lexical tree search based decoder). Thus, the method greatly simplifies ASR building process with even smaller amounts of coding. Currently, it took 7-9 days by using a single GPU to train the network with full training data (581h) in the CSJ task, which is comparable to the whole training time of the conventional state-of-the-art system due to the simplification of building process.

Future work will apply this technique to the other languages including English, where we have to solve an issue of long sequence lengths, which requires heavy computation cost and makes it difficult to train a decoder network. Actually, neural machine translation handles this issue by using a sub word unit (concatenating several letters to form a new sub word unit) (Wu et al., 2016), which would be a promising direction for end-to-end ASR.

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

## A Examples of irregular alignments

We list examples of irregular alignments caused by attention-based ASR. Figure 3 shows an example of repetitions of word chunks. The first chunk of blue characters in attention-based ASR is appeared again, and the whole second chunk part becomes insertion errors. Figure 4 shows an example of deletion errors. The latter half of the sentence in attention-based ASR is broken, which causes deletion errors. The proposed joint CTC-attention avoids these issues.

Figure 3: Example of insertion errors appeared in attention-based ASR.

Figure 4: Example of deletion errors appeared in attention-based ASR.

## B Location-based attention mechanism

This section provides the equations of a location-based attention mechanism $\text{Attention}(\cdot)$ in Eq. (10).

$$a_{lt} = \text{Attention}(\{a_{l-1}\}_t, \mathbf{q}_{l-1}, \mathbf{h}_t),$$

where $\{a_{l-1}\}_t = [a_{l-1,1}, \cdots, a_{l-1,T}]^\top$. To obtain $a_{lt}$, we use the following equations:

$$\{\mathbf{f}_{lt}\}_t = \mathbf{K} * \mathbf{a}_{l-1} \tag{21}$$

$$e_{lt} = \mathbf{g}^\top \tanh(\mathbf{G}^{\text{q}}\mathbf{q}_{l-1} + \mathbf{G}^{\text{h}}\mathbf{h}_t + \mathbf{G}^{\text{f}}\mathbf{f}_{lt} + \mathbf{b}) \tag{22}$$

$$a_{lt} = \frac{\exp(e_{tl})}{\sum_t \exp(e_{tl})} \tag{23}$$

$\mathbf{K}, \mathbf{G}^{\text{q}}, \mathbf{G}^{\text{h}}, \mathbf{G}^{\text{f}}$ are matrix parameters. $\mathbf{b}$ and $\mathbf{g}$ are vector parameters. $*$ denotes convolution along input frame axis $t$ with $\mathbf{K}$.

