# Peer review of "Joint CTC/attention decoding for end-to-end speech recognition"

_ACL 2017 — decision unknown_

[Official Review · Reviewer 1 · rating 3 · confidence 4]
soundness 5 · originality 5 · clarity 4 · impact 3 · substance 4 · appropriateness 4 · meaningful comparison 3 · presentation format Poster

This paper proposes joint CTC-attention end-to-end ASR, which utilizes both
advantages in training and decoding. 

- Strengths:
It provides a solid work of hybrid CTC-attention framework in training and
decoding, and the experimental results showed that the proposed method could
provide an improvement in Japanese CSJ and Mandarin Chinese telephone speech
recognition task. 

- Weaknesses:
The only problem is that the paper sounds too similar with Ref [Kim et al.,
2016] which will be officially published in the coming IEEE International
Conference on Acoustics, Speech, and Signal Processing (ICASSP), March 2017.
Kim at al., 2016, proposes joint CTC-attention using MTL for English ASR task,
and this paper proposes joint CTC-attention using MTL+joint decoding for
Japanese and Chinese ASR tasks. I guess the difference is on joint decoding and
the application to Japanese/Chinese ASR tasks. However, the difference is not
clearly explained by the authors. So it took sometimes to figure out the
original contribution of this paper.

(a) Title: 
The title in Ref [Kim et al., 2016] is “Joint CTC- Attention Based End-to-End
Speech Recognition Using Multi-task Learning”, while the title of this paper
is “Joint CTC-attention End-to-end Speech Recognition”. I think the title
is too general. If this is the first paper about "Joint CTC-attention" than it
is absolutely OK. Or if Ref [Kim et al., 2016] will remain only as
pre-published arXiv, then it might be still acceptable. But since [Kim et al.,
2016] will officially publish in IEEE conference, much earlier than this paper,
then a more specified title that represents the main contribution of this paper
in contrast with the existing publication would be necessary. 

(b) Introduction:
The author claims that “We propose to take advantage of the constrained CTC
alignment in a hybrid CTC-attention based system. During training, we attach a
CTC objective to an attention-based encoder network as a regularization, as
proposed by [Kim at al., 2016].“ Taking advantage of the constrained CTC
alignment in a hybrid CTC-attention is the original idea from [Kim at al.,
2016]. So the whole argument about attention-based end-to-end ASR versus
CTC-based ASR, and the necessary of CTC-attention combination is not novel.
Furthermore, the statement “we propose … as proposed by [Kim et al,
2016]” is somewhat weird. We can build upon someone proposal with additional
extensions, but not just re-propose other people's proposal. Therefore, what
would be important here is to state clearly the original contribution of this
paper and the position of the proposed method with respect to existing
literature

(c) Experimental Results:
Kim at al., 2016 applied the proposed method on English task, while this paper
applied the proposed method on Japanese and Mandarin Chinese tasks. I think it
would be interesting if the paper could explain in more details about the
specific problems in Japanese and Mandarin Chinese tasks that may not appear in
English task. For example, how the system could address multiple possible
outputs. i.e., Kanji, Hiragana, and Katakana given Japanese speech input
without using any linguistic resources. This could be one of the important
contributions from this paper.

- General Discussion:
I think it would be better to cite Ref [Kim et al., 2016] from
the official IEEE ICASSP conference, rather than pre-published arXiv:
Kim, S., Hori, T., Watanabe, S., "Joint CTC- Attention Based End-to-End Speech
Recognition Using Multi-task Learning", IEEE International Conference on
Acoustics, Speech, and Signal Processing (ICASSP), March 2017, pp. to appear.

[Official Review · Reviewer 2 · rating 3 · confidence 5]
soundness 5 · originality 5 · clarity 4 · impact 3 · substance 4 · appropriateness 5 · meaningful comparison 3 · presentation format Poster

The paper considers a synergistic combination of two non-HMM based speech
recognition techniques: CTC and attention-based seq2seq networks. The
combination is two-fold:
1. first, similarly to Kim et al. 2016 multitask learning is used to train a
model with a joint CTC and seq2seq cost.
2. second (novel contribution), the scores of the CTC model and seq2seq model
are ensembled during decoding (results of beam search over the seq2seq model
are rescored with the CTC model).

The main novelty of the paper is in using the CTC model not only as an
auxiliary training objective (originally proposed by Kim et al. 2016), but also
during decoding.

- Strengths:
The paper identifies several problems stemming from the flexibility offered by
the attention mechanism and shows that by combining the seq2seq network with
CTC the problems are mitigated.

- Weaknesses:
The paper is an incremental improvement over Kim et al. 2016 (since two models
are trained, their outputs can just as well be ensembled). However, it is nice
to see that such a simple change offers important performance improvements of
ASR systems.

- General Discussion:
A lot of the paper is spent on explaining the well-known, classical ASR
systems. A description of the core improvement of the paper (better decoding
algorithm) starts to appear only on p. 5. 

The description of CTC is nonstandard and maybe should either be presented in a
more standard way, or the explanation should be expanded. Typically, the
relation p(C|Z) (eq. 5) is deterministic - there is one and only one character
sequence that corresponds to the blank-expanded form Z. I am also unsure about
the last transformation of the eq. 5.